

# A JavaScript API for the Ice Sheet System Model: towards on online interactive model for the Cryosphere Community

Eric Larour[1], Daniel Cheng[2], Gilberto Perez[2], Justin Quinn[2], Mathieu Morlighem[3], Bao Duong[4], Lan Nguyen[5], Kit Petrie[1], Silva Harounian[6], Daria Halkides[7], and Wayne Hayes[2]

[1]Jet Propulsion Laboratory - California Institute of technology, 4800 Oak Grove Drive MS 300-323, Pasadena, CA 91109-8099, USA
[2]University of California, Irvine, Department of Information and Computer Sciences, Donald Bren Hall, Irvine, CA 92697-3100, USA
[3]University of California, Irvine, Department of Earth System Science, Croul Hall, Ivine, CA 92697-3100, USA
[4]Monoprice, Inc.11701 6th Street Rancho Cucamonga, CA 91730, USA.
[5]Hart, Inc., 1515 E Orangewoord ave Anaheim, CA 92805 USA
[6]Digitized Schematic Solutions, Address: 40 W. Cochran st. Suite: 212 Simi Valley CA 93065, California, USA
[7] Earth and Space Research, 2101 Fourth Ave., Suite 1310, Seattle, WA 98121, USA.

*Correspondence to:* Eric Larour (eric.larour@jpl.nasa.gov)

**Abstract.**

Earth System Models (ESMs) are becoming increasingly complex, requiring extensive knowledge and experience to deploy and use in an efficient manner. They run on high-performance architectures that are significantly different from the everyday environments that scientists use to pre and

post-process results (i.e. MATLAB, Python). This results in models that are hard to use for non specialists, and that are increasingly specific in their application. It also makes them relatively inaccessible to the wider science community, not to mention to the general public. Here, we present a new software/model paradigm that attempts to bridge the gap between the science community and the complexity of ESMs, by developing a new JavaScript Application Program Interface (API) for

the Ice Sheet System Model (ISSM). The aforementioned API allows Cryosphere Scientists to run ISSM on the client-side of a webpage, within the JavaScript environment. When combined with a Web server running ISSM (using a Python API), it enables the serving of ISSM computations in an easy and straightforward way. The deep integration and similarities between all the APIs in ISSM (MATLAB, Python, and now JavaScript) significantly shortens and simplifies the turnaround

of state-of-the-art science runs and their use by the larger community. We demonstrate our approach via a new Virtual Earth System Laboratory (VESL) Web site.



## 1  Introduction

Earth System Models (ESMs) across the Earth science community have become increasingly so-
phisticated, enabling more accurate simulations and projections of the Earth's climate as well as the
state of the atmosphere, ocean, land, ice, and biosphere. As demonstrated by the Coupled Model In-
tercomparison Project 5 (CMIP-5, Taylor et al., 2009, 2012) and its new iteration (CMIP-6, Eyring
et al., 2016) of the World Climate Research Programme (WCRP), the multiplicity of ESMs, and the
complexity of the physics they capture, is significant. The description of the outputs for CMIP-5
runs is 133 pages long by itself, showing the complexity and comprehensive nature of the processes
modeled in the ESMs that participated in the project. Any one of these models is massive both in
terms of the number of lines of code, but also in terms of structure and modularity (or lack thereof).
GEOS-5 for example (Molod et al., 2015), one of the Atmosphere and Ocean General Circulation
Models (AOGCMs) that participated in CMIP-5, is made of 600,000 lines of Fortran code, compris-
ing 88 physical modules (as of Jan 2016). This is fairly representative of the complexity of ESMs
nowadays, and of the multiplicity of physical processes necessary to realistically model the evolution
of the whole Earth System.

The above described complexity results in serious issues regarding the way simulations are run.
For example, what we generally define as pre-processing and post-processing phases are increas-
ingly different from the computational phase itself. The computational core is usually written in C
or Fortran, which easily supports parallelism and High Performance Computing (HPC). However, in
the pre-processing phase, where datasets are processed into a binary file used by the computational
core, or in the post-processing phase, where simulation results are visualized, scientific environments
such as MATLAB or Python are increasingly relied upon. Indeed, presently, these environments are
almost entirely ubiquitous within the science community. This results in additional complexity to
manage different environments: scientists are well-acquainted with the difficulties of porting their
software to HPC instances, while struggling to process the data inputs and results on local worksta-
tions where data upload/download can be a limiting factor, hard drive memory requirements substan-
tial, and problems due to the use of different APIs significant (MATLAB, Python, and IDL, among
others).

Another complexity originating from the wide variety of physical processes represented in ESMs
is the difficulty in initializing a computational run. For example, in the Ice Sheet System Model
(ISSM, Larour et al., 2012), one of the land ice components of GEOS-5, developed at the National
Aeronautics and Space Administration (NASA) Jet Propulsion Laboratory (JPL), in collaboration
with University of California, Irvine (UCI), the initialization setup for the Greenland Ice Sheet (GIS)
transient simulations from 1850 to present day amounts to 3,000 lines of MATLAB code. This com-
prises model setup, data interpolation onto an ISSM compatible mesh, solution parameterizations,
and initialization strategies, among other things. This simulation, part of the Ice Sheet Modeling
Intercomparison Project 6 (ISMIP-6 Nowicki et al., 2016) that accounts for ice sheets in CMIP-6,



is a fairly representative example of some of the most advanced simulations that can be run with

an Ice Sheet Model (ISM). Such simulations cannot easily be systematized and need to be tailored specifically for each ice sheet they are applied to. Our JavaScript API, however, allows a scientist to handle sophisticated computations without having to become an expert in each and every one of the software modules required by the physics of a given simulation.

One of the approaches that could mitigate some of the issues discussed above involves the devel-

opment of computational frameworks capable of serving ESM simulations. This type of solution involves running simulations that already include pre and post-processing phases (i.e.where the model setup has already been carried out or is carried out by the server itself by uploading key datasets) and in which the user is allowed to control only a few, key parameters. Similarly, once the computation is carried out on the server-side, the results are post-processed automatically, and only significant

results are provided to the user. This type of approach has already been explored, for example, in areas relating to serving of large datasets, such as the NASA Earth Observing System Data and Information System (EOSDIS) EarthData server, which provides a portal with integrated processing capabilities for large scale datasets collected by NASA missions. However, fewer examples of this kind of approach are available that serve simulation results, and to our knowledge, no comprehen-

sive ESM, nor module thereof, has ever been integrated into a server solution capable of delivering ESM computations on the fly. The reason for this is simple: the complexity of the physics involved is significant, reconciling pre/post processing phases and simulation cores is inherently difficult, and basing a simulation framework on server technologies represents a significant software development challenge.

Specifically, the bottlenecks that preclude deeper integration of ESMs within server infrastructures include: 1) Bridging the gap between ESM formulations of the physical cores and Web technologies such as HyperText Markup Language (HTML, World Wide Web Consortium, 1997) and JavaScript (ECMA International, 2016), which are not languages that are inherently used by Earth scientists, for reasons that are obvious. Because ESMs are not natively integrated into Web technologies, it renders

the link between server infrastructures and simulation engines difficult; 2) The significant turnaround between generation and serving of simulations. This lag is due to the fact that these two processes are inherently different in the way they are designed and, moreover, are usually considered to be completely separate phases of what should, essentially, be the same process; 3) The distributed nature of Web simulations. Every step of an ESM run can be considered a separate, logical component. For

example, post-processing of a simulation may be done on a different machine than the one that initially generated it; and 4) The lack of existing integrated frameworks wherein simulations, pre and post-processing, and the serving of the data and/or simulation results all occur within the same architecture.

Here, we present a new approach applied to the ISSM framework, a land ice model of signif-

icant size and complexity, to serve simulation results relating to the evolution of polar ice sheets.



Our solution is based on a new JavaScript API for the ISSM framework itself, allowing it to be fully integrated within a Web environment, namely an HTML webpage (described in Section 2). By leveraging its existing Python API within a FastCGI module of an Apache HTTP server, we show (in Section 3) how ISSM can be used to provide simulations and to pre and post-process results

directly. This new approach allows for a quick turnaround between model setup and simulation, in particular shortening the time required to carry out science runs and to serve the results to the wider science community (Section 4). We execute this approach (Section 5) within the newly-designed Virtual Earth System Laboratory (VESL), demonstrating how we can provide cryosphere-related simulations to the science community, and to the wider public in general, thereby easily providing

access to the wide array of modular physics embedded in ISSM. We conclude with a discussion of the potential of this new approach to both facilitate a wider use of ESMs by scientists of varied disciplines and to shorten the gap between science and public outreach.

## 2    ISSM JavaScript API

Most ISMs are written in Fortran, C, or C++, for reasons related to computational efficiency and to

the ease of integration within HPC environments using parallel libraries, such as Message Passing Interface (MPI) via OpenMP (Gropp et al., 1996; Gropp and Lusk, 1996; OpenMP Architecture Review Board, 2015). However, many simpler models exist that rely on different APIs, such as the MATLAB code described in MacAyeal (1993) or the Excel-based Greenland and Antarctica Ice Sheet Model designed for educational purposes (GRANTISM, Pattyn, 2005). These models have

in common the desire to rely on a simple code base, and to reduce/optimize the set of physics captured in the code, in order to make it more accessible. Our approach here, however, is to facilitate accessibility without sacrificing the complexity and full-set of features of ISSM by implementing a brand new API using the JavaScript language. The goal is to be able to integrate ISSM within Web-based solutions, relying on JavaScript as a language that enables control of the behavior of an HTML

webpage. In addition, by making the JavaScript API similar in all possible aspects to the existing MATLAB and Python ISSM APIs, model runs and simulations can be transferred easily to the Web, furthering our objective of disseminating ISSM to the larger scientific community and, possibly, the general public.

    The basis for representing a model in ISSM is a series of classes (mesh, mask, geometry, settings,

toolkits, etc.) that are carried into a global `model` class. The first task was, therefore, to translate all ISSM classes from MATLAB and Python into JavaScript. Fig. 1 shows an example of such a translation for the `mesh2d` class (used to represent a 2D mesh triangulation comprising a list of vertex coordinates x,y of size `numberofvertices` with corresponding `lat,long` coordinates, a list of triangle indices called `elements` (of size `numberofelements`), and a projection code

using an EPSG Geodetic Parameter Dataset). The constructors are very similar, and there is a one-





to-one correspondence between the mesh2d methods in both APIs. The example of the `marshall` routine (which collects all the mesh info onto a binary buffer that will be sent to the ISSM C++ core) shows the similarity between both codes, with differences in the syntax reduced to a bare minimum. This equivalence is essential in preserving all of the physics captured in each class of ISSM, and

could only be achieved because MATLAB, Python, and JavaScript are similar in their syntax and grammar.

The `savemodel` routine in the MATLAB `mesh2d` class implementation is unique, as it allows users to run simulations in MATLAB using ISSM, and, once the simulations are over, to save the MATLAB defined model into a JavaScript equivalent file. This routine, which closely matches the

constructor, is the key to shortening the transition time between the setup of an ISSM simulation and its transition to a webpage environment. The fact that all of the information of a given class is identical in both APIs demonstrates the comprehensiveness of the new JavaScript implementation of ISSM, and that it achieves its goal of replicating ISSM within a webpage environment.

In addition to the classes representation in JavaScript, ISSM relies on a C++ core for computations.

This C++ code is present at several levels: 1) For each pre and post-processing module (or, wrapper) that requires significant computational power, such as interpolation routines that transfer information between gridded dataset and unstructured Finite Element Modeling (FEM) meshes typical of ISSM; and 2) For each of the computations pertaining to ice flow itself (the physical engine in ISSM), which we refer to as the ISSM core. For both sets of code, a solution relying on the Emscripten

compiler (Zakai, 2011) was deployed. This compiler enables translation of C++ code directly into JavaScript, with computational efficiencies that are within an order of magnitude of the translated C++ code. Listing 1 shows how Emscripten was integrated within the existing Makefile structure of ISSM. All the pre and post-processing wrappers (TriMesh, NodeConnectivity, ContourToMesh, ElementConnectivity, InterpFromMeshToMesh2d, IssmConfig, EnumToString, and StringToEnum)

as well as the ISSM core itself (issm) are compiled into JavaScript executables using the C++ files and a set of Emscripten related flags (described in the IssmModule_CXXFLAGS variable). This Makefile is similar to its MATLAB and Python counterparts, with the exception of the issm core, which is compiled as a JavaScript module instead of a C++ executable. This Makefile is integrated within Autotools (Vaughan et al., 2000), enabling for quick activation of the compilation using a

simple "`--with-javascript`" option during the configuration phase of the ISSM software.

The JavaScript modules and ISSM core are continuously tested against regression tests, similar to the MATLAB and Python APIs (Larour et al., 2012). The integration framework for the tests relies on Jenkins, an open-source automation server (Jenkins, 2016), which provides continuous integration and delivery of validated ISSM code. The ISSM Jenkins webpage is available at https:

//ross.ics.uci.edu:8080/, where the entire validation suite is in the process of being transferred to JavaScript. This ensures that continuous development impacts all of the APIs in ISSM in a similar





fashion without imparting delays to the JavaScript API (due to the fact that it would be used by a smaller base of ISSM users).

## 3    HTTP/Python Server

Using the JavaScript API, it is possible to run a full-fledged simulation using any of the physical modules described in Larour et al. (2012). However, to our knowledge, Emscripten does not yet allow computations in parallel within a browser. This limits the range of model sizes and mesh resolution to a level that compromises large-scale simulations. In these cases, our approach was to rely on the cloud computing capabilities of ISSM, as described in Larour and Schlegel (2016), and to

host a Web server that would deliver ISSM computations to any client running the ISSM JavaScript API. This server relies on the Python API of ISSM to carry out computations ranging from tens to hundreds of thousands of degrees of freedom, allowing continental-scale simulations. The server is fully-elastic and scalable, and relies on the Amazon EC2 infrastructure (Amazon, 2008), and can spin-up Compute Optimized CC4.8x large instances (up to 64 threads of computational power) on

demand, making it a robust solution for serving computations. Refer to Larour and Schlegel (2016) for more details on this part of the architecture.

In terms of server configuration itself, our approach was to rely on the Python API of ISSM to leverage the FastCGI Web interface, described in Market (1996), on an Apache server. This allows requests coming into the Apache server from the client-side to be routed directly to a Python script.

The Web client, running ISSM embedded inside JavaScript, can therefore upload a marshalled binary input file (created by the call to the marshall routine of each model class, as described in Fig. 1) to the EC2 instance Apache server, which then routes it to the Python script that launches the parallel job.

Fig. 2 describes this process schematically, and compares it to what happens in more classic

simulations relying on MATLAB and an HPC infrastructure, such as a cluster. The fundamental differences between the traditional simulation paradigm and our new solution are: 1) The client architecture, which runs either MATLAB or an HTML webpage with JavaScript; 2) The upload/download of binary input files, which is done either through an SSH copy call or an XMLHttpRequest, respective to the aforementioned client architectures; and 3) The launching of a given computation,

which is handled via a queuing system on the head node or a FastCGI-relayed Python call on an EC2 instance, again, respective to the client architecture. In terms of parallel computations, ISSM executables are run using an MPI call in both cases. The strong similarity between both architectures was purposefully designed so as to limit the amount of repeated code, and to ensure the robustness of the computations themselves, which are transparent to the API they rely upon.



## 4 All-In-One Design/Simulations

Listing 2 shows a typical model setup for a simulation in ISSM relying on the MATLAB API. The steps include loading a model (or generating one using a mesher), modifying a certain input parameter, such as surface mass balance (SMB), setting up a cluster class (pointing to the parallel cluster) and calling the solve routine. Once the results are carried out/downloaded, plotmodel is run to visualize them.

An additional step can be carried out once a given MATLAB ISSM model has been built, wherein the model is saved into a JavaScript file (md.js) in some webpage directory. This model can then be used (as shown in Listing. 3) to run the exact same setup and simulation as is done with MATLAB, but on the client's machine. The HTML code for this simulation is typical of a webpage, and includes: 1) Standard HTML markup (i.e. W3C-compliant html, head, and body objects); 2) Include statements for the ISSM binaries created by Emscripten, the model itself (md.js), and a sort of front-end controller (engine.js, which controls the display of and interaction with the simulation on the webpage); and 3) HTML elements such as a canvas where the results will be plotted (similar to the figure statement in MATLAB), a second canvas for the color bar, and a button element to launch the simulation. The listing for engine.js shows how similar the MATLAB and JavaScript setup are. Upon loading, if the RUN button is clicked, the value of a slider (SMB value) is retrieved and then SolveGlacier called. The SolveGlacier() routine modifies the SMB parameter, sets up the cluster class (pointing to the EC2 server), and calls the solve routine. After computations are carried out and downloaded, a callback function PlotGlacier is triggered, which plots the model results onto the aforementioned HTML canvas elements.

Fig. 3 shows an example of such a webpage hosting a simulation on the impact of anomalies in SMB on transient ice flow (including Mass Transport and Stress Balance) on the Columbia Glacier in Alaska. This webpage is part of VESL, where the JavaScript API of ISSM was leveraged along with the HTTP/Python Server architecture described previously to showcase the capabilities of ISSM to serve computations on the fly and to visualize them instantly (Larour et al., 2016). The simulations within VESL are all simulations that were carried out using ISSM for scientific publications. By adding a savemodeljs step at the end of the MATLAB simulation workflow, and by replicating some of that same workflow in the engine.js code, it is possible to deploy a simulation like the one described above on a Web platform with significantly shortened turnaround and using the exact same capabilities as the initial MATLAB solution itself. This breakthrough is only possible because of the duplication of the entire architecture: again, by making JavaScript code that is logically equivalent to our MATLAB or Python constructs and by mapping the whole workflow described in Fig. 2 from MATLAB/HPC infrastructures to HTML/JavaScript/EC2. Our methodology paves the way to leveraging Web technologies and cloud computing to host large-scale simulations of modeling engines such as ISSM, all without loss of the physical representation of processes nor scalability.





## 5 Examples

Fig. 3 and Fig. 4 show examples of simulations that rely on the ISSM JavaScript API, and that are hosted on the VESL Web site (Larour et al., 2016). VESL's purpose is to twofold: to showcase simulations that demonstrate ISSM capabilities, and to demonstrate the capabilities of our new Web-based modeling solution to the wider scientific community and general public. Several simulations are hosted, leveraging the large set of capabilities in ISSM.

The first simulations pertain to the simulation of glacier flow, mainly from work on Haig Glacier (Adhikari and Marshall, 2011) and Columbia Glacier (Gardner, Fahnestock, Larour, pers. comm.). Fig. 3 shows the Columbia Glacier webpage is, where SMB anomalies (to the background trend) can be specified, with ISSM then computing the resulting modification to the transient flow over a period of 10 years. This simulation includes mass transport, stress balance, a basal friction parameter, which was inverted using surface velocities (Fahnestock, Gardner and Larour, pers. comm.) following Morlighem et al. (2013), and thermal steady-state.

The second set of simulations pertain to ice sheet modeling in Antarctica and Greenland. Fig. 4 shows the webpage corresponding to the friction SeaRISE (Bindschadler et al., 2013) experiment over the entire Greenland ice sheet. In this simulation, which is also calculated using a basal friction inversion from surface velocities (Rignot, 2008), thermal steady-state, and stress-balance (no mass transport, nor transient), the user can decrease the friction under the ice at the ice/bedrock interface and compute the resulting changes in surface velocities. The model is fairly high resolution (12,000 elements), which allows for physically-relevant computations.

The third set of simulations pertains to Sea-Level Rise (SLR) modeling, relying on the ISSM-SESAW module (Adhikari et al., 2016) to compute gravitationally consistent sea-level and geodetic signatures caused by cryosphere and climate-driven mass change. Presently, two sets of simulations demonstrate: 1) Eustatic SLR and its impact on coastline migration in the USA; and 2) SLR from eustatic, gravity, and elastic deformation on a global scale, wherein users can turn off specific sets of SLR physics to understand the impact of gravitation on redistribution of SLR around the world and the impact of local elastic deformation of the Earth lithosphere.

Finally, a fourth set of simulations pertains to Solid Earth deformation, using the ISSM-GIA (Adhikari and Marshall, 2011) module that captures Glacial Isostatic Adjustment (GIA) from ice-sheet loading. It should be noted that this section is a work in progress.

One potential future section may feature recent work by the ISSM team involving the application of ISSM to other planets (namely, Mars' ice caps). Given the relatively quick turnaround between ISSM simulations and their porting to the Web using the ISSM JavaScript API, our hope is that VESL will become a forum for cryosphere scientists to discuss ice sheet related science. In addition, by enabling simplified interfaces on existing simulations that resulted in scientific publications, we believe the general public might gain increased interest in this type of approach to better understand the complexities of science for the Earth system as a whole.



## 6 Conclusions

We developed a fully-functional JavaScript API for the Ice Sheet System Model (ISSM), which allows cryosphere scientists to carry out ice flow simulations within a Web environment. This API gives access to the entire spectrum of physical processes captured by ISSM without compromising its complexity and richness. For simulations requiring parallel computing, the JavaScript API can be leveraged against a computational server hosted on a cloud instance (such as Amazon EC2) to deliver high-performance, large-scale, and high-fidelity simulations back to the Web client. This new set of capabilities enables hosting of high-end simulations on the NASA/JPL ESL, effectively solving a fundamental challenge of ESMs: delivering accessible, high-performance simulations in a timely manner is historically and inherently difficult. We believe that our approach paves the way for the efficient deployment of feature-rich ESM's, a quick turnaround between scientific work and corresponding publications, and outreach not only the science community but also to the general public.

## 7 Code Availability

The ISSM code and its JS components are available at http://issm.jpl.nasa.gov. The instructions for the compilation of ISSM in JS mode is presented in the supplement attached to this manuscript.

*Acknowledgements.* This work was performed at the Jet Propulsion Laboratory (JPL), California Institute of Technology, and the Department of Earth System Science at the University of California, Irvine (UCI) under a contract with the National Aeronautics and Space Administration (NASA) and funded by the Cryospheric Sciences Program. Resources supporting the numerical simulations were provided by the NASA High-End Computing (HEC) Program through the NASA Advanced Supercomputing (NAS) Division at Ames Research Center, and by Cryospheric Program Management for the Amazon EC2 instances hosting the Virtual Earth System Laboratory. We would like to thank Dr. Alex Gardner from JPL and Dr. Mark Fahnestock from the University of Alaska, Fairbanks for the datasets used in the Columbia Glacier model setup of the Virtual Earth System Laboratory. We would also like to thank Dr. Daisy F. Sang for the supervision of students from CalPoly Pomona who participated in this project over the past 5 years.

**Listing 1.** Makefile for Javascript Emscripten compilation of ISSM.

```
EXEEXT=js

js_scripts = ${ISSM_DIR}/src/wrappers/TriMesh/TriMesh.js   \
${ISSM_DIR}/src/wrappers/NodeConnectivity/NodeConnectivity.js \
${ISSM_DIR}/src/wrappers/ContourToMesh/ContourToMesh.js \
${ISSM_DIR}/src/wrappers/ElementConnectivity/ElementConnectivity.js \
${ISSM_DIR}/src/wrappers/InterpFromMeshToMesh2d/InterpFromMeshToMesh2d.js \
${ISSM_DIR}/src/wrappers/IssmConfig/IssmConfig.js \
```





```
       ${ISSM_DIR}/src/wrappers/EnumToString/EnumToString.js \
       ${ISSM_DIR}/src/wrappers/StringToEnum/StringToEnum.js \
       ${ISSM_DIR}/src/wrappers/Issm/issm.js
       bin_SCRIPTS  = issm−prebin.js
       bin_PROGRAMS = IssmModule

       issm−prebin.js: ${js_scripts}
cat ${js_scripts}  > issm−prebin.js

       IssmModule_SOURCES = ../TriMesh/TriMesh.cpp \
                    ../NodeConnectivity/NodeConnectivity.cpp\
                    ../ContourToMesh/ContourToMesh.cpp\
315                 ../ElementConnectivity/ElementConnectivity.cpp\
                    ../InterpFromMeshToMesh2d/InterpFromMeshToMesh2d.cpp\
                    ../IssmConfig/IssmConfig.cpp\
                    ../EnumToString/EnumToString.cpp\
                    ../StringToEnum/StringToEnum.cpp\
320                 ../Issm/issm.cpp

       IssmModule_CXXFLAGS= −fPIC −D_DO_NOT_LOAD_GLOBALS_  −−memory−init−file 0 \
       $(AM_CXXFLAGS) $(CXXFLAGS) $(CXXOPTFLAGS) $(COPTFLAGS) \
       −s EXPORTED_FUNCTIONS="['_TriMeshModule','_NodeConnectivityModule',\
'_ContourToMeshModule','_ElementConnectivityModule',\
       '_InterpFromMeshToMesh2dModule','_IssmConfigModule','_EnumToStringModule'\
       ,'_StringToEnumModule','_IssmModule']"  −s DISABLE_EXCEPTION_CATCHING=0 \
       −s ALLOW_MEMORY_GROWTH=1 −s INVOKE_RUN=0

IssmModule_LDADD = ${deps} $(TRIANGLELIB)  $(GSLLIB)
```





```
%MESH2D class definition
classdef mesh2d
    properties (SetAccess=public)
        x                       = NaN;
        y                       = NaN;
        elements                = NaN;
        numberofelements        = 0;
        numberofvertices        = 0;
        numberofedges           = 0;
        lat                     = NaN;
        long                    = NaN;
        epsg                    = 0;
    end
    methods
+-- 18 lines: function self = mesh2d(varargin) % ------------------------------
+--  9 lines: function self = setdefaultparameters(self) % --------------------
+-- 19 lines: function md = checkconsistency(self,md,solution,analyses) % -----
function marshall(self,md,fid) % {{{
WriteData(fid,'enum',DomainTypeEnum(),'data',StringToEnum(['Domain' domaintype(self)]),'format','Integer');
WriteData(fid,'enum',DomainDimensionEnum(),'data',dimension(self),'format','Integer');
WriteData(fid,'enum',MeshElementtypeEnum(),'data',StringToEnum(elementtype(self)),'format','Integer');
WriteData(fid,'object',self,'class','mesh','fieldname','x','format','DoubleMat','mattype',1);
WriteData(fid,'object',self,'class','mesh','fieldname','y','format','DoubleMat','mattype',1);
WriteData(fid,'enum',MeshZEnum(),'data',zeros(self.numberofvertices,1),'format','DoubleMat','mattype',1);
WriteData(fid,'object',self,'class','mesh','fieldname','elements','format','DoubleMat','mattype',2);
WriteData(fid,'object',self,'class','mesh','fieldname','numberofelements','format','Integer');
WriteData(fid,'object',self,'class','mesh','fieldname','numberofvertices','format','Integer');
end % }}}
+--  3 lines: function t = domaintype(self) % ---------------------------------
+--  3 lines: function d = dimension(self) % ----------------------------------
+--  3 lines: function s = elementtype(self) % --------------------------------
function savemodeljs(self,fid,modelname) % {{{
writejs1Darray(fid,[modelname '.mesh.x'],self.x);
writejs1Darray(fid,[modelname '.mesh.y'],self.y);
writejs2Darray(fid,[modelname '.mesh.elements'],self.elements);
writejsdouble(fid,[modelname '.mesh.numberofelements'],self.numberofelements);
writejsdouble(fid,[modelname '.mesh.numberofvertices'],self.numberofvertices);
writejsdouble(fid,[modelname '.mesh.numberofedges'],self.numberofedges);
writejs1Darray(fid,[modelname '.mesh.lat'],self.lat);
writejs1Darray(fid,[modelname '.mesh.long'],self.long);
writejsdouble(fid,[modelname '.mesh.epsg'],self.epsg);
end % }}}
end
end
```

```
//MESH2D class definition
function mesh2d () {
    //methods
+-- 10 lines: this.setdefaultparameters = function (){ --------------------------
+--  3 lines: this.classname = function () { -----------------------------------
+--  3 lines: this.domaintype=function (){ -------------------------------------
+--  3 lines: this.dimension = function () { -----------------------------------
+--  3 lines: this.elementtype = function() {-----------------------------------
this.marshall=function(md,fid) { //{{{
WriteData(fid,'enum',DomainTypeEnum(),'data',StringToEnum('Domain' + this.domaintype()),'format','Integer');
WriteData(fid,'enum',DomainDimensionEnum(),'data',this.dimension(),'format','Integer');
WriteData(fid,'enum',MeshElementtypeEnum(),'data',StringToEnum(this.elementtype()),'format','Integer');
WriteData(fid,'object',this,'class','mesh','fieldname','x','format','DoubleMat','mattype',1);
WriteData(fid,'object',this,'class','mesh','fieldname','y','format','DoubleMat','mattype',1);
WriteData(fid,'enum',MeshZEnum(),'data',NewArrayFill(this.numberofvertices,0),'format','DoubleMat','mattype',1)
WriteData(fid,'object',this,'class','mesh','fieldname','elements','format','DoubleMat','mattype',2);
WriteData(fid,'object',this,'class','mesh','fieldname','numberofelements','format','Integer');
WriteData(fid,'object',this,'class','mesh','fieldname','numberofvertices','format','Integer');
}//}}}
+-- 12 lines: this.fix=function() { ---------------------------------------------
    //properties
    // {{{
    this.x                      = NaN;
    this.y                      = NaN;
    this.elements               = NaN;
    this.numberofelements       = 0;
    this.numberofvertices       = 0;
    this.numberofedges          = 0;

    this.lat                    = NaN;
    this.long                   = NaN;
    this.epsg                   = 0;

    this.setdefaultparameters();
    //}}}
}
```

**Figure 1.** Line by line comparison of the code behind the mesh2d class, within the MATLAB ISSM API (upper frame) and the JavaScript ISSM API (lower frame). Routines followed by a dashed line have been folded for ease of reading.





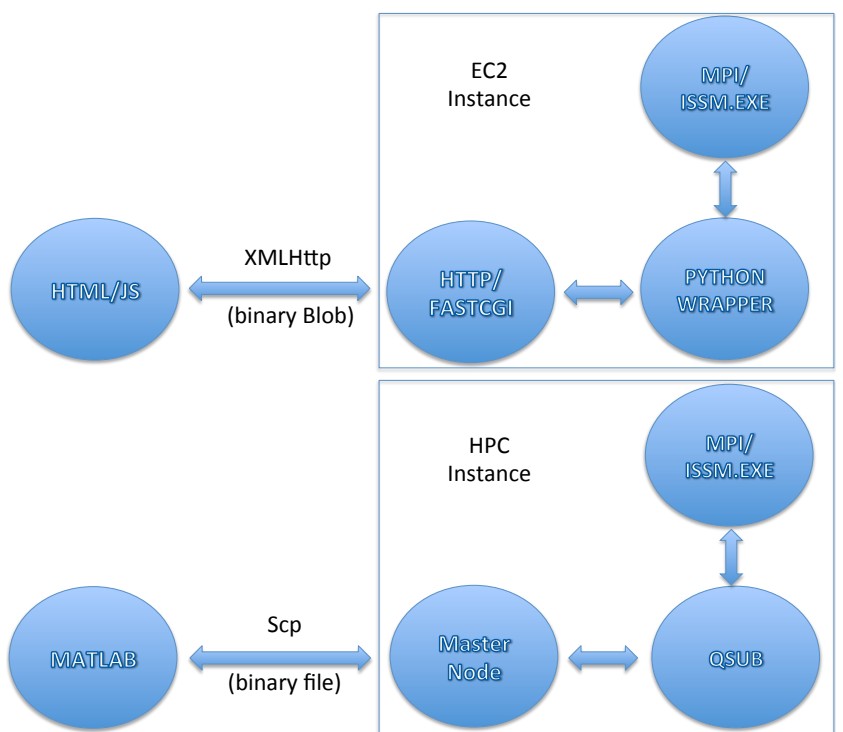

**Figure 2.** Similarities between a standard MATLAB/HPC ISSM run and a JavaScript/EC2 driven run. In the first case (lower frames) a MATLAB instance running on a local workstation running the ISSM API marshalls an input binary file, which is then uploaded (using an ssh call) to a master node on a cluster. The binary file is then queued into the system (using a qsub command, for example). The parallel runs are then carried out using the ISSM executable and an MPI-compatible environment. In the second case, a browser client makes an XMLHttpRequest and uploads a Binary Large Object (the exact same binary file MATLAB would upload), which is received by an HTTP server (e.g. Apache) running on an Amazon EC2 compute-optimized instance. The HTTP server then uses a FastCGI module to interface to a Python wrapper, which automatically triggers a system call to the MPI environment running the ISSM executable. In both cases, an output binary file is created by the ISSM executable, which is then shipped back to the MATLAB instance or the client's Web browser.

---

```
    ;
```

**Listing 2.** MATLAB code for a typical simulation of the Virtual Earth System Laboratory (VESL).

```
% Load Model :
md=loadmodel ( 'Models/md.mat ' ) ;

% Solve :
```





```
    md.smb.mass_balance= smb_initial;
    for i=1:md.mesh.numberofvertices,
      md.smb.mass_balance(i) = md.smb.mass_balance(i)+ smbvalue;
    end
    md.cluster=generic('name','localhost','np',8);
    md=solve(md,TransientSolutionEnum());

    %Plot Model:
vel=md.results.TransientSolution(1).Vel;
    plotmodel(md,'data',vel,'log',10,'figure',1,'colorbar','on',...
          'overlay','on','images','radar.png');

    % Export to JS model:
md.savemodeljs('md',websiteroot);
```

**Listing 3.** Equivalent (see Listing. 2) Hmtl/Javascript code for a typical simulation within the Virtual Earth System Laboratory. Prototype webpage.

```
<html>
  <script type="text/javascript" src="./bin/issm-binaries.js"></script>
  <script type="text/javascript" src="./src/engine.js"></script>
  <script type="text/javascript" src="./js/md.js"></script>
  <body data-spy="scroll" data-target="nav" onload="engine();">

    <canvas id="columbia"></canvas>

<div class="bordered margin-8 padding-8">
        <canvas id="columbia-colorbar" class="colorbar-v"> </canvas>
      </div>

      <div id="columbia-run" class="bordered margin-8 padding-8">
365       <button type="button" class="interactive run-button"
          onclick="SolveGlacier()"> RUN </button>
      </div>
</html>
function engine(){
  PlotGlacier();
  slider('value',0,'callback',function(value){smbvalue=value;},
```

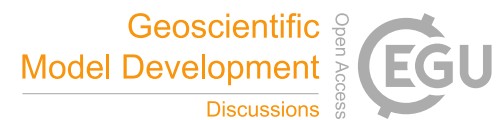



```
        'name','columbia', 'min',-5,'max',+5,'message',['SMB anomaly:','m/a'],
'step',.1, 'slidersdiv','columbia-sliders');
    }

    function SolveGlacier(){
      md.smb.mass_balance= smb_initial.slice(0);
for (var i=0;i<md.mesh.numberofvertices;i++){
        md.smb.mass_balance[i] += smbvalue;
      }
      md.cluster=new generic('url',server + '/fastcgi/issm_solve.py','np',8);
      md=solve(md,TransientSolutionEnum(),'checkconsistency','no',
'callback',PlotGlacier);
    }

    function PlotGlacier(){
      plotmodel(md,'data',md.results[0]['Vel'],'log',10,'canvasid#all','columbia',
390       'colorbar','on', 'colorbarcanvasid','columbia-colorbar',
        'overlay','on','image', './images/radar.png');
    }
```





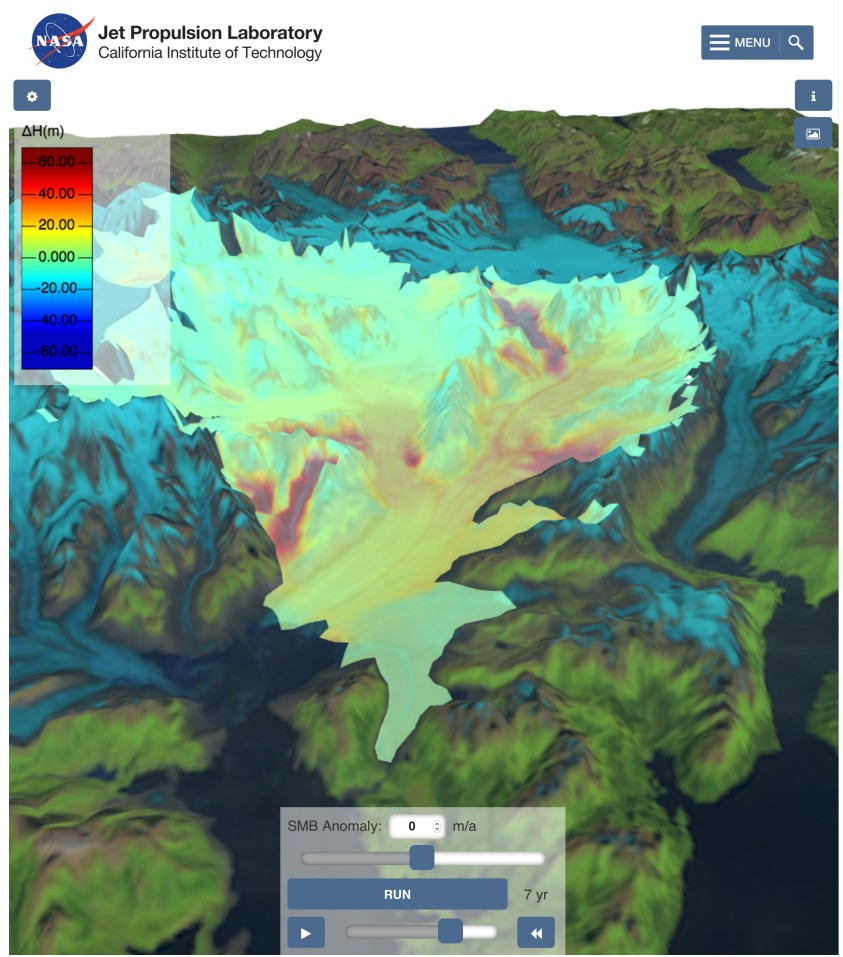

**Figure 3.** Columbia Glacier ISSM simulation on the Virtual Earth System Laboratory (http://issm.jpl.nasa.gov/ earthsystemlaboratorynew). This particular simulation allows for the introduction of user-driven SMB anomalies (using a slider ranging from -5 to +5 m/a) on the transient ice flow of Columbia Glacier. The computations (upon clicking of the RUN button) are carried out on the ISSM computational server (where the model inputs are uploaded, and from which the results are downloaded locally to the client's Web browser). The transient results are displayed as a movie, which can be controlled via user interface (UI) controls. The interactive rendering of the velocity and thickness fields is done in 3D (or 2D, upon clicking of a toggle button) using the ISSM WebGL rendering engine. The results are overlaid on a semi-transparent topographical rendering of the SRTM DEM, and a background geotiff image from Gardner et al (pers. comm.). Model information can be displayed by clicking the info button, allowing for extensive information on the model setup and the datasets used to constrain the simulation.

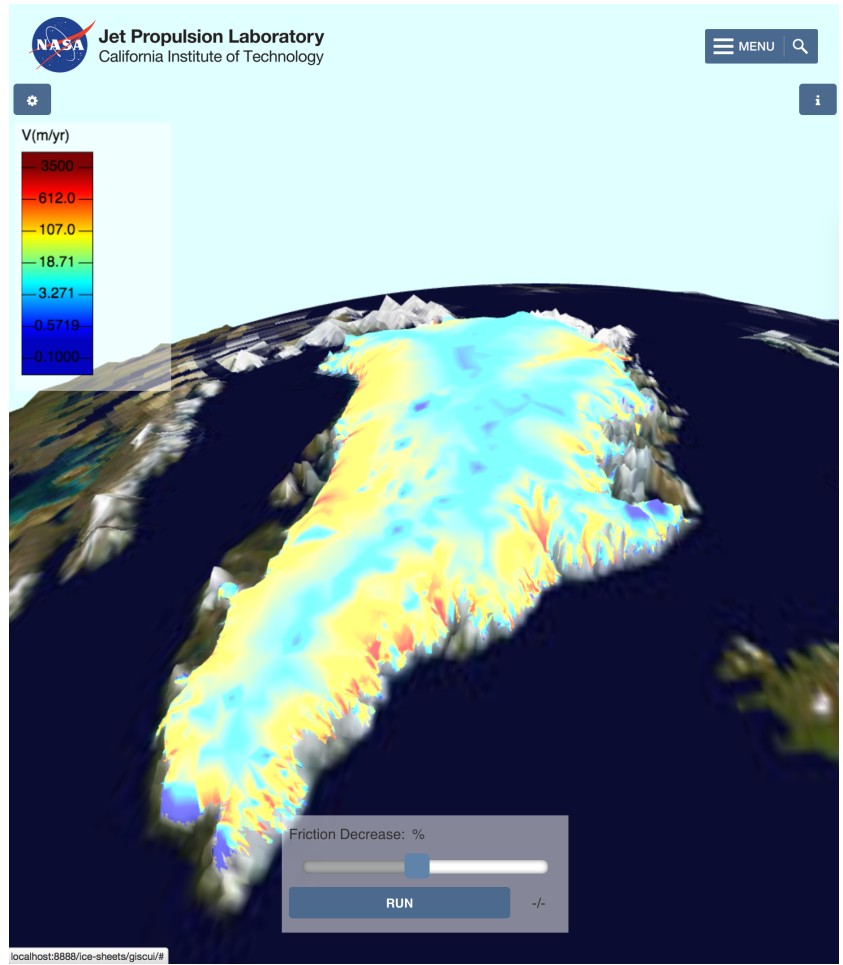

**Figure 4.** Greenland ISSM simulation on the Virtual Earth System Laboratory (ESL) (http://issm.jpl.nasa.gov/
earthsystemlaboratorynew). This particular simulation allows for the introduction of user-driven friction anoma-
lies (using a slider ranging from 5 to 100%) on the steady-state stress-balance velocities for the entire Greenland
Ice Sheet. The computations (upon clicking of the RUN button) are carried out on the ISSM computational
server (where the model inputs are uploaded, and from which the results are downloaded locally to the client'
Web browser). The steady-state velocities are displayed for each value of the friction coefficient that the user
chooses. The interactive rendering of the velocity field is done in 3D using the ISSM WebGl rendering engine.
The results are overlaid on a semi-transparent topographical rendering of ETOPO5 data (see reference: National
Geophysical Data Center (1988) for credits) and a background geotiff image from the Blue Marble: Land Sur-
face, Shallow Water and Shaded Topography project (see reference: NASA Goddard Space Flight Center, Reto
Stockli for credits).



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
