# Peer review of "A JavaScript API for the Ice Sheet System Model (ISSM) 4.11: towards an online interactive model for the Cryosphere Community"

_Geoscientific Model Development, 2016_

## Short Comment (SC1) · 13 Sep 2016

Dear authors,

In my role as Executive editor of GMD, I would like to bring to your attention our Editorial version 1.1:

http://www.geosci-model-dev.net/8/3487/2015/gmd-8-3487-2015.html

This highlights some requirements of papers published in GMD, which is also available on the GMD website in the 'Manuscript Types' section:

http://www.geoscientific-model-development.net/submission/manuscript_types.html

In particular, please note that for your paper, the following requirements have not been met in the Discussions paper:

- "The main paper must give the model name and version number (or other unique identifier) in the title."

- "If the model development relates to a single model then the model name and the version number must be included in the title of the paper. If the main intention of an article is to make a general (i.e. model independent) statement about the usefulness of a new development, but the usefulness is shown with the help of one specific model, the model name and version number must be stated in the title. The title could have a form such as, "Title outlining amazing generic advance: a case study with Model XXX (version Y)"."

In order to simplify reference to your developments, please add a version number and consider to add the models name acronym in the title of your article in your revised submission to GMD.

Yours,

Astrid Kerkweg

———————————————————

---

## Referee Comment (RC1) · Anonymous Referee #1 · 7 Oct 2016

General comments: This paper works to integrate the Ice Sheet System Model (ISSM) into an interactive online model that can be accessed by a larger community. It transforms a series of existing MATLAB and Python classes, currently used by ISSM users to generate model runs, into equivalent JavaScript classes, enabling direct deployment in a web environment. It leverages commercial cloud virtual machine and web service technologies to enable rapid generation of ISSM results via user adjustments of model parameters within a web browser.

This work represents an important step forward in bringing the powerful capabilities of Earth System Models to a larger community. Conventionally, a researcher interested in exploring such a model must invest considerable time to learn FORTRAN or C++

codes, or if they are lucky, there are MATLAB or Python scripts that are provided as wrappers around the lower level codes. Nevertheless, users must still invest much time to handle protocols surrounding input data formatting, methods for parameter adjustment, and other methods for controlling the model. Deploying such a model in a web environment will enable a wide new range of model exploration to take place.

Specific comments:

Para beginning line 132: This paragraph is unclear. Is "savemodel" unique because it occurs only in the MATLAB implementation? Is this a mechanism for visualizing local simulations set up using MATLAB? Please clarify.

Lines 139-150: Until this section, I understood that existing MATLAB and Python pre-configuration wrappers were being converted to JavaScript, but not the main ISSM code. However, this section discusses converting everything to JavaScript. Later, it becomes clearer that the core ISSM code is used in a parallel configuration on EC2 for the larger, continental scale model runs (presumably the C++ version?). I suggest setting this up more clearly earlier in the manuscript, explaining the two primary implementations and the reasons for each approach. Presumably the EC2 simulations would involve a user setting a simulation to run and the results being returned after some time? How would that be handled in the web environment (e.g. the user is emailed when results are ready?).

Line 217: on the assumption that the audience may not know a lot of glaciology, I suggest explaining SMB, transient ice flow, etc in more general terms (e.g. surface climate forcings, etc).

Line 222: this is providing a bit more clarity on the savemodel component, but I am still not discovering the "breakthrough" it enables. Is there some kind of caching of initial output being done here to speed up the implementation in the web browser?

Fig 3 caption: fix: ". . .shows the Columbia Glacier webpage is, . . .". Again, quite a bit

[Figure]

of glaciology jargon here.

Fig 2: It is good to see a conceptual map like this, but some additional detail could help. The labeling has considerable amount of acronyms. A few simple terms identifying that the client is on the left, the server on the right, and the flow of input/output through the diagram, would help, especially for those not immediately aware of all the different terms.
* * *

---

## Referee Comment (RC2) · Anonymous Referee #2 · 5 Jan 2017

General comments

The paper presents the development of a Javascript Interface for the Ice Sheet System Model, aimed at simplifying the interaction and running of the model for less specialist users. Making modelling and the results of modelling more accessible is key to aiding wider understanding of the research. The paper is largely well written, therefore, I recommend it for publication, however, there are a few things that could be clarified.

The main thing that isn't clear to me is who the API is targeted at – on page 4 it reads as if the ISSM experts will still be setting up the model runs, and the API is mainly a tool for communicating the results of the modelling to the wider community and possibly the public. In the examples you have in the VESL, these are relatively simple (I realise they

are demonstrators) and all the scenarios (e.g. SMB change) could be pre-run and the API is then just a tool to allow the user to engage with the results, it doesn't need the model to be re-run by every user, this seems to be a waste of computational resource. Even in more complex scenarios, it still seems like pre-running the simulations and making just the results available for interrogation would be more efficient. But - is this what is happening – is this what is meant on line 223? Are the results on the VESL all pre-run? This really isn't clear to me.

So how accessible is this API for a non-specialist to use to set up their own domain, and to set up the web server, for example on the Amazon EC2 infrastructure? And what is the cost? This is not clear to me from the paper as it is written, and leaves me in doubt as to whether the API would simplify things for a less experienced user to set up their own domain, and even whether this is one of the intentions for the API.

I think what I would like to see is a clearer statement of the purpose and advantages for different users, perhaps the information is there, but I got to the end of the paper not entirely the wiser.

Specific comments

Lines 75-85: This numbered list is hard to follow because some of your points are long and contain full stops, I forgot what the list was about by the time I hit point 2). I suggest adding line breaks before each point, making them more like a bullet pointed list.

Line 79: Nothing should be considered as obvious, please remove the statement, or elaborate on the reasons!

Lines 132 onwards: as with reviewer 1, I got a bit confused in this bit as to what was written in Matlab and what was in C++, please clarify

Line 215: If the user wants to present the results in a different way, is it possible for them to extract the data from the API, or do they have to use the inbuilt visualisation options?

Line 239: remove the "is" after webpage

---

## Author Comment (AC1) · 28 Jan 2017

**1 Editor Comment #1**

Dear authors,

In my role as Executive editor of GMD, I would like to bring to your attention our Editorial version 1.1:

http://www.geosci-model-dev.net/8/3487/2015/gmd-8-3487-2015.html
This highlights some requirements of papers published in GMD, which is also
available on the GMD website in the Manuscript Types section:
http://www.geoscientific-model-development.net/submission/manuscript\_types.html
In particular, please note that for your paper, the following requirement has not
been met in the Discussions paper:

- "The main paper must give the model name and version number (or other unique identifier) in the title."
- If the model development relates to a single model then the model name and the version number must be included in the title of the paper. If the main intention of an article is to make a general (i.e. model independent) statement about the usefulness of a new development, but the usefulness is shown with the help of one specific model, the model name and version number must be stated in the title. The title could have a form such as, Title outlining amazing generic advance: a case study with Model XXX (version Y).

In order to simplify reference to your developments, please add a version number and consider to add the models name acronym in the title of your article in your revised submission to GMD. Yours,

Astrid Kerkweg

We thank the executive editor for catching this issue, and have accordingly added the ISSM version number to the title along with the acronym for ISSM.

**2 Reviewer #1:**

General comments:

This paper works to integrate the Ice Sheet System Model (ISSM) into an interactive online model that can be accessed by a larger community. It transforms a series of existing MATLAB and Python classes, currently used by ISSM users to generate model runs, into equivalent JavaScript classes, enabling direct deployment in a web environment. It leverages commercial cloud virtual machine and web service technologies to enable rapid generation of ISSM results via user adjustments of model parameters within a web browser. This work represents an important step forward in bringing the powerful capabilities of Earth System Models to a larger community. Conventionally, a researcher interested in exploring such a model must invest considerable time to learn FORTRAN or C++ codes, or if they are lucky, there are MATLAB or Python scripts that are provided as wrappers around the lower level codes. Nevertheless, users must still invest much time to handle protocols surrounding input data formatting, methods for parameter adjust- ment, and other methods for controlling the model. Deploying such a model in a web environment will enable a wide new range of model exploration to take place.

We thank the reviewer for the positive assessment of the manuscript, and for the time spent giving advice on how to improve the flow of the text. We have tried to improve the presentation of the new methods, to better frame for which audience they are relevant, and for what type of computational scenarios the new JavaScript API is used. This was a concern echoed also by reviewer 2. Modifications were therefore made in particular to the introduction.

Specific comments:

Para beginning line 132: this paragraph is unclear. Is savemodel unique because it occurs only in the MATLAB implementation? Is this a mechanism for visualizing local simulations set up using MATLAB? Please clarify.

We have reformulated this paragraph to explain better the philosophy behind the "savemodel" routine, the fact it's unique to MATLAB, and allows to shorten the turn-around between running a MATLAB based scientic simulation in ISSM, and transferring its results, along with the underlying "model" class, to a webpage using a JavaScript include file.

Lines 139-150: Until this section, I understood that existing MATLAB and Python pre- configuration wrappers were being converted to JavaScript, but not the main ISSM code. However, this section discusses converting everything to JavaScript. Later, it becomes clearer that the core ISSM code is used in a parallel configuration on EC2 for the larger, continental scale model runs (presumably the C++ version?). I suggest setting this up more clearly earlier in the manuscript, explaining the two primary im- plementations and the reasons for each approach. Presumably the EC2 simulations would involve a user setting a simulation to run and the results being returned after some time? How would that be handled in the web environment (e.g. the user is emailed when results are ready?).

We understand the confusion, and have strived to clarify this point (see reviewer 2's remark on the subject too) in the paragraph by reworking this aspect extensively. We now clearly differentiate the C++ core (used for large-scale model runs on the Amazon EC2 instances) and the JavaScript core (equivalent of the C++ core but translated into JS using emscripten), used for computations local to the webpage that do not require parallel computing (small models). As

suggested, we also introduce this point earlier on in the introduction.

Line 217: on the assumption that the audience may not know a lot of glaciology, I suggest explaining SMB, transient ice flow, etc in more general terms (e.g. surface climate forcings, etc).

We took the reviewer's advice and simplified the text to make it more accessibility to a larger audience.

Line 222: this is providing a bit more clarity on the savemodel component, but I am still not discovering the breakthrough it enables. Is there some kind of caching of initial output being done here to speed up the implementation in the web browser?

We have tried to clarify this statement. The breakthrough is really in shortening the turn around between running a science study with ISSM, and getting it transferred to a JavaScript environment. It's not related at all with caching results, as also suggested by reviewer 2, but just deals litterally with sending the model object to a new environment, a WebPage. We tried and modify the manuscript accordingly.

Fig 3 caption: fix: . . .shows the Columbia Glacier webpage is, . . .. Again, quite a bit of glaciology jargon here.

Fixed the typo and simplified/improved the text further.

Fig 2: It is good to see a conceptual map like this, but some additional detail could help. The labeling has considerable amount of acronyms. A few simple terms identifying that the client is on the left, the server on the right, and the flow of input/output through the diagram, would help, especially for those not immediately aware of all the different terms.

We have considerably improved the figure by taking the reviewer's advice, and recasting the caption to introduce the relevant acronyms instead of having them on the figure itself. We generalized the terminology to make it more accessible to a general audience.

**3 Reviewer #2**

General comments:

The paper presents the development of a Javascript Interface for the Ice Sheet System Model, aimed at simplifying the interaction and running of the model for less specialist users. Making modelling and the results of modelling more accessible is key to aiding wider understanding of the research. The paper is largely well written, therefore, I recommend it for publication, however, there are a few things that could be clarified.

The main thing that isnt clear to me is who the API is targeted at on page 4 it reads as if the ISSM experts will still be setting up the model runs, and the API is mainly a tool for communicating the results of the modelling to the wider community and possibly the public. In the examples you have in the VESL, these are relatively simple (I realise they are demonstrators) and all the scenarios (e.g. SMB change) could be pre-run and the API is then just a tool to allow the user to engage with the results, it doesn't need the model to be re-run by every user, this seems to be a waste of computational resource. Even in more complex scenarios, it still seems like pre-running the simulations and making just the results available for interrogation would be more efficient. But - is this what is happening is this what is meant on line 223? Are the results on the VESL all pre-run? This really isn't clear to me.

We thank the reviewer for the effort in the review, the acknowledgement of the importance of making modeling and the results of modeling more accessible to a wider audience. We realize that somehow, we missed some clear statements in the text as to who the audience was, what the method for releasing results would be, and whether this would actually make things easier for new users, or whether this was directed at experts in ISSM who want to release professional quality results to a larger audience. We have accordingly corrected the introduction of the manuscript to make this abundantly clear. We would like to respectfully push back on the statement that the models presented in VESL are relatively simple. These are reseach-grade simulations, with resolutions and simulation times difficult to achieve without the use of a cluster. The SeaRISE experiment presented in Figure 4 was litterally taken from the ISSM SeaRISE runs carried out in 2014. This goes to a deeper misunderstanding that we tried to clarify in the manuscript, that originated from the savemodel routine description, in which both reviewers thought that we were caching the results. It is absolutely not the intent of this new API to do so, as we believe that having ISSM run live during a simulation is a key feature. We do not agree with the reviewer that this would be a waste of computational ressources, as this would be an exciting opportunity to showcase science capabiliities at their maximum potential, without degrading their quality, which is usually the first casualty of outreach and education. We again make this point clear in the text, and why we believe this is the way outreach should go.

So how accessible is this API for a non-specialist to use to set up their own domain, and to set up the web server, for example on the Amazon EC2 infrastructure? And what is the cost? This is not clear to me from the paper as it is written, and leaves me in doubt as to whether the API would simplify things for a less experienced user to set up their own domain, and even whether this is one of the intentions for the API. This relates to the point raised before by the reviewer. The tool is not intended to facilitate use of ISSM per se, but its integration in a new environment that is web based. We are currently working on a full-fledged ISSM simulation portal on a webpage, but most of the time, this API will be used in conjunction with an already setup model from a scientist (from pre-existing Matlab or Python runs) to enable quicker dissemination of the results, and replication of the simulation on a webpage. This will not make things easier in terms of modeling, but it will make things easier on the scientist who will not need to be an expert in web design to on his own transfer his result and simulation engine to a webpage. Again, we have modified the introduction in this respect to make these points clear.

I think what I would like to see is a clearer statement of the purpose and advantages for different users, perhaps the information is there, but I got to the end of the paper not entirely the wiser.

**See above.**

Specific comments:

Lines 75-85: This numbered list is hard to follow because some of your points are long and contain full stops, I forgot what the list was about by the time I hit point 2). I suggest adding line breaks before each point, making them more like a bullet pointed list.

We actually took the reviewer's advice litterally, and went for an enumeration list, as the four points presented here are significant enough.

Line 79: Nothing should be considered as obvious, please remove the statement, or elaborate on the reasons!

Thank you for the advice, we have rephrased the statement accordingly to further explain why we believe HTML and JS are languages not used by the scientific community.

Lines 132 onwards: as with reviewer 1, I got a bit confused in this bit as to what was written in Matlab and what was in C++, please clarify.

The paragraph has been reworked (see equivalent remark from reviewer 1) to clearly differentiate the C++ core (used for large-scale model runs on the Amazon EC2 instances) and the JavaScript core (equivalent of the C++ core but translated into JS using emscripten), used for computations local to the web-page that do not require parallel computing (small models).

Line 215: If the user wants to present the results in a different way, is it possible

for them to extract the data from the API, or do they have to use the inbuilt visualisation options?

This is a very interesting question. The results are provided directly by the call to the "solve" routine (see Listing 3) in engine.js. The user is therefore free to modify engine.js to use his own rendering engine directly. We have hinted at this aspect in the manuscript.

Line 239: remove the is after webpage

Done.

Manuscript prepared for Geosci. Model Dev. with version 2015/04/24 7.83 Copernicus papers of the LATEX class copernicus.cls. Date: 27 January 2017

**A JavaScript API for the Ice Sheet System Model (ISSM) 4.11: towards an online interactive model for the Cryosphere Community**

Eric Larour1, Daniel Cheng2, Gilberto Perez2, Justin Quinn2, Mathieu Morlighem3, Bao Duong4, Lan Nguyen5, Kit Petrie1, Silva Harounian6, Daria Halkides7, and Wayne Hayes2

[revised manuscript text omitted]

| %MESH2D class definition                                                                                                                                                                                                                                                                                                                                                                                                                                                                                                                                                                                                                                                                                                                                                                                                                                                                                                                                                                                                                                                                                                                                                                                                                                                                                                                                                                                                                                                                                                                                                                                                                                                                                                                                                                                                                                                                                                                                                                                                                                                                                                                                                                                                                                                                                                                                                                                         |                                                                                                                                                     |
|------------------------------------------------------------------------------------------------------------------------------------------------------------------------------------------------------------------------------------------------------------------------------------------------------------------------------------------------------------------------------------------------------------------------------------------------------------------------------------------------------------------------------------------------------------------------------------------------------------------------------------------------------------------------------------------------------------------------------------------------------------------------------------------------------------------------------------------------------------------------------------------------------------------------------------------------------------------------------------------------------------------------------------------------------------------------------------------------------------------------------------------------------------------------------------------------------------------------------------------------------------------------------------------------------------------------------------------------------------------------------------------------------------------------------------------------------------------------------------------------------------------------------------------------------------------------------------------------------------------------------------------------------------------------------------------------------------------------------------------------------------------------------------------------------------------------------------------------------------------------------------------------------------------------------------------------------------------------------------------------------------------------------------------------------------------------------------------------------------------------------------------------------------------------------------------------------------------------------------------------------------------------------------------------------------------------------------------------------------------------------------------------------------------|-----------------------------------------------------------------------------------------------------------------------------------------------------|
| classdef mesh2d                                                                                                                                                                                                                                                                                                                                                                                                                                                                                                                                                                                                                                                                                                                                                                                                                                                                                                                                                                                                                                                                                                                                                                                                                                                                                                                                                                                                                                                                                                                                                                                                                                                                                                                                                                                                                                                                                                                                                                                                                                                                                                                                                                                                                                                                                                                                                                                                  |                                                                                                                                                     |
| X                                                                                                                                                                                                                                                                                                                                                                                                                                                                                                                                                                                                                                                                                                                                                                                                                                                                                                                                                                                                                                                                                                                                                                                                                                                                                                                                                                                                                                                                                                                                                                                                                                                                                                                                                                                                                                                                                                                                                                                                                                                                                                                                                                                                                                                                                                                                                                                                                | = NaN;                                                                                                                                              |
| У                                                                                                                                                                                                                                                                                                                                                                                                                                                                                                                                                                                                                                                                                                                                                                                                                                                                                                                                                                                                                                                                                                                                                                                                                                                                                                                                                                                                                                                                                                                                                                                                                                                                                                                                                                                                                                                                                                                                                                                                                                                                                                                                                                                                                                                                                                                                                                                                                | = NaN;                                                                                                                                              |
| elements
numberofelements                                                                                                                                                                                                                                                                                                                                                                                                                                                                                                                                                                                                                                                                                                                                                                                                                                                                                                                                                                                                                                                                                                                                                                                                                                                                                                                                                                                                                                                                                                                                                                                                                                                                                                                                                                                                                                                                                                                                                                                                                                                                                                                                                                                                                                                                                                                                                                                     | = NaN;
= 0                                                                                                                                       |
| numberofvertices                                                                                                                                                                                                                                                                                                                                                                                                                                                                                                                                                                                                                                                                                                                                                                                                                                                                                                                                                                                                                                                                                                                                                                                                                                                                                                                                                                                                                                                                                                                                                                                                                                                                                                                                                                                                                                                                                                                                                                                                                                                                                                                                                                                                                                                                                                                                                                                                 | = 0;
= 0;                                                                                                                                        |
| numberofedges                                                                                                                                                                                                                                                                                                                                                                                                                                                                                                                                                                                                                                                                                                                                                                                                                                                                                                                                                                                                                                                                                                                                                                                                                                                                                                                                                                                                                                                                                                                                                                                                                                                                                                                                                                                                                                                                                                                                                                                                                                                                                                                                                                                                                                                                                                                                                                                                    | = 0;                                                                                                                                                |
| lat                                                                                                                                                                                                                                                                                                                                                                                                                                                                                                                                                                                                                                                                                                                                                                                                                                                                                                                                                                                                                                                                                                                                                                                                                                                                                                                                                                                                                                                                                                                                                                                                                                                                                                                                                                                                                                                                                                                                                                                                                                                                                                                                                                                                                                                                                                                                                                                                              | = NaN;                                                                                                                                              |
| epsa                                                                                                                                                                                                                                                                                                                                                                                                                                                                                                                                                                                                                                                                                                                                                                                                                                                                                                                                                                                                                                                                                                                                                                                                                                                                                                                                                                                                                                                                                                                                                                                                                                                                                                                                                                                                                                                                                                                                                                                                                                                                                                                                                                                                                                                                                                                                                                                                             | = NCN;
= 0:                                                                                                                                      |
| end                                                                                                                                                                                                                                                                                                                                                                                                                                                                                                                                                                                                                                                                                                                                                                                                                                                                                                                                                                                                                                                                                                                                                                                                                                                                                                                                                                                                                                                                                                                                                                                                                                                                                                                                                                                                                                                                                                                                                                                                                                                                                                                                                                                                                                                                                                                                                                                                              | -,                                                                                                                                                  |
| methods                                                                                                                                                                                                                                                                                                                                                                                                                                                                                                                                                                                                                                                                                                                                                                                                                                                                                                                                                                                                                                                                                                                                                                                                                                                                                                                                                                                                                                                                                                                                                                                                                                                                                                                                                                                                                                                                                                                                                                                                                                                                                                                                                                                                                                                                                                                                                                                                          |                                                                                                                                                     |
| + 18 lines: function self = mesh2
+ 9 lines: function self = setde                                                                                                                                                                                                                                                                                                                                                                                                                                                                                                                                                                                                                                                                                                                                                                                                                                                                                                                                                                                                                                                                                                                                                                                                                                                                                                                                                                                                                                                                                                                                                                                                                                                                                                                                                                                                                                                                                                                                                                                                                                                                                                                                                                                                                                                                                                                                            | d(varargin) %                                                                                                                                       |
| + 19 lines: function md = checkco                                                                                                                                                                                                                                                                                                                                                                                                                                                                                                                                                                                                                                                                                                                                                                                                                                                                                                                                                                                                                                                                                                                                                                                                                                                                                                                                                                                                                                                                                                                                                                                                                                                                                                                                                                                                                                                                                                                                                                                                                                                                                                                                                                                                                                                                                                                                                                                | nsistency(self,md,solution,analyses) %                                                                                                              |
| <pre>function marshall(self,md,fid) % {{</pre>                                                                                                                                                                                                                                                                                                                                                                                                                                                                                                                                                                                                                                                                                                                                                                                                                                                                                                                                                                                                                                                                                                                                                                                                                                                                                                                                                                                                                                                                                                                                                                                                                                                                                                                                                                                                                                                                                                                                                                                                                                                                                                                                                                                                                                                                                                                                                                   | {                                                                                                                                                   |
| WriteData(fid, 'enum', DomainTypeEnum                                                                                                                                                                                                                                                                                                                                                                                                                                                                                                                                                                                                                                                                                                                                                                                                                                                                                                                                                                                                                                                                                                                                                                                                                                                                                                                                                                                                                                                                                                                                                                                                                                                                                                                                                                                                                                                                                                                                                                                                                                                                                                                                                                                                                                                                                                                                                                            | <pre>(),'data',StringToEnum(['Domain' domaintype(self)]),'format','Integer');
nEnum() 'data' dimension(self) 'format' 'Integen');</pre>         |
| WriteData(fid, 'enum', MeshElementtyp                                                                                                                                                                                                                                                                                                                                                                                                                                                                                                                                                                                                                                                                                                                                                                                                                                                                                                                                                                                                                                                                                                                                                                                                                                                                                                                                                                                                                                                                                                                                                                                                                                                                                                                                                                                                                                                                                                                                                                                                                                                                                                                                                                                                                                                                                                                                                                            | eEnum(), 'data', StringToEnum(elementtype(self)), 'format', 'Integer');                                                                             |
| WriteData(fid, 'object', self, 'class'                                                                                                                                                                                                                                                                                                                                                                                                                                                                                                                                                                                                                                                                                                                                                                                                                                                                                                                                                                                                                                                                                                                                                                                                                                                                                                                                                                                                                                                                                                                                                                                                                                                                                                                                                                                                                                                                                                                                                                                                                                                                                                                                                                                                                                                                                                                                                                           | <pre>,'mesh','fieldname','x','format','DoubleMat','mattype',1);</pre>                                                                               |
| WriteData(fid, 'object', self, 'class'                                                                                                                                                                                                                                                                                                                                                                                                                                                                                                                                                                                                                                                                                                                                                                                                                                                                                                                                                                                                                                                                                                                                                                                                                                                                                                                                                                                                                                                                                                                                                                                                                                                                                                                                                                                                                                                                                                                                                                                                                                                                                                                                                                                                                                                                                                                                                                           | ,'mesh','fieldname','y','format','DoubleMat','mattype',1);                                                                                          |
| <pre>WriteData(fid 'object' self 'class' WriteData(fid 'object' self 'class'</pre>                                                                                                                                                                                                                                                                                                                                                                                                                                                                                                                                                                                                                                                                                                                                                                                                                                                                                                                                                                                                                                                                                                                                                                                                                                                                                                                                                                                                                                                                                                                                                                                                                                                                                                                                                                                                                                                                                                                                                                                                                                                                                                                                                                                                                                                                                                                               | ata', zeros (self.numberofvertices,1),'format','DoubleMat','mattype',1);
'mesh' 'fieldname' 'elements' 'format' 'DoubleMat' 'mattype' 2); |
| WriteData(fid, 'object', self, 'class'                                                                                                                                                                                                                                                                                                                                                                                                                                                                                                                                                                                                                                                                                                                                                                                                                                                                                                                                                                                                                                                                                                                                                                                                                                                                                                                                                                                                                                                                                                                                                                                                                                                                                                                                                                                                                                                                                                                                                                                                                                                                                                                                                                                                                                                                                                                                                                           | <pre>, 'mesh', 'fieldname', 'numberofelements', 'format', 'Integer');</pre>                                                                         |
| WriteData(fid, 'object', self, 'class'                                                                                                                                                                                                                                                                                                                                                                                                                                                                                                                                                                                                                                                                                                                                                                                                                                                                                                                                                                                                                                                                                                                                                                                                                                                                                                                                                                                                                                                                                                                                                                                                                                                                                                                                                                                                                                                                                                                                                                                                                                                                                                                                                                                                                                                                                                                                                                           | ,'mesh','fieldname','numberofvertices','format','Integer');                                                                                         |
| end % }}                                                                                                                                                                                                                                                                                                                                                                                                                                                                                                                                                                                                                                                                                                                                                                                                                                                                                                                                                                                                                                                                                                                                                                                                                                                                                                                                                                                                                                                                                                                                                                                                                                                                                                                                                                                                                                                                                                                                                                                                                                                                                                                                                                                                                                                                                                                                                                                                         | mo(colf) K                                                                                                                                          |
| + 3 lines: function $t = domainty$
+ 3 lines: function $d = dimension$                                                                                                                                                                                                                                                                                                                                                                                                                                                                                                                                                                                                                                                                                                                                                                                                                                                                                                                                                                                                                                                                                                                                                                                                                                                                                                                                                                                                                                                                                                                                                                                                                                                                                                                                                                                                                                                                                                                                                                                                                                                                                                                                                                                                                                                                                                                                        | n(self) %                                                                                                                                           |
| + 3 lines: function s = elementt                                                                                                                                                                                                                                                                                                                                                                                                                                                                                                                                                                                                                                                                                                                                                                                                                                                                                                                                                                                                                                                                                                                                                                                                                                                                                                                                                                                                                                                                                                                                                                                                                                                                                                                                                                                                                                                                                                                                                                                                                                                                                                                                                                                                                                                                                                                                                                                 | ype(self) %                                                                                                                                         |
| <pre>function savemodeljs(self,fid,model</pre>                                                                                                                                                                                                                                                                                                                                                                                                                                                                                                                                                                                                                                                                                                                                                                                                                                                                                                                                                                                                                                                                                                                                                                                                                                                                                                                                                                                                                                                                                                                                                                                                                                                                                                                                                                                                                                                                                                                                                                                                                                                                                                                                                                                                                                                                                                                                                                   | name) % {{{                                                                                                                                         |
| <pre>writejs1Darray(fid, _modelname '.mes writejs1Darray(fid _modelname '.mes</pre>                                                                                                                                                                                                                                                                                                                                                                                                                                                                                                                                                                                                                                                                                                                                                                                                                                                                                                                                                                                                                                                                                                                                                                                                                                                                                                                                                                                                                                                                                                                                                                                                                                                                                                                                                                                                                                                                                                                                                                                                                                                                                                                                                                                                                                                                                                                              | h.x'],self.x);
h.v'] self.v);                                                                                                                    |
| writejs2Darray(fid, [modelname '.mes                                                                                                                                                                                                                                                                                                                                                                                                                                                                                                                                                                                                                                                                                                                                                                                                                                                                                                                                                                                                                                                                                                                                                                                                                                                                                                                                                                                                                                                                                                                                                                                                                                                                                                                                                                                                                                                                                                                                                                                                                                                                                                                                                                                                                                                                                                                                                                             | h.elements'],self.elements);                                                                                                                        |
| writejsdouble(fid, [modelname '.mesh                                                                                                                                                                                                                                                                                                                                                                                                                                                                                                                                                                                                                                                                                                                                                                                                                                                                                                                                                                                                                                                                                                                                                                                                                                                                                                                                                                                                                                                                                                                                                                                                                                                                                                                                                                                                                                                                                                                                                                                                                                                                                                                                                                                                                                                                                                                                                                             | .numberofelements'],self.numberofelements);                                                                                                         |
| writejsdouble(fid, [modelname '.mesh                                                                                                                                                                                                                                                                                                                                                                                                                                                                                                                                                                                                                                                                                                                                                                                                                                                                                                                                                                                                                                                                                                                                                                                                                                                                                                                                                                                                                                                                                                                                                                                                                                                                                                                                                                                                                                                                                                                                                                                                                                                                                                                                                                                                                                                                                                                                                                             | <pre>.numberofvertices'],self.numberofvertices);</pre>                                                                                              |
| writejsdouble(fid,[modelname '.mesh
writejs1Darrav(fid,[modelname '.mesh                                                                                                                                                                                                                                                                                                                                                                                                                                                                                                                                                                                                                                                                                                                                                                                                                                                                                                                                                                                                                                                                                                                                                                                                                                                                                                                                                                                                                                                                                                                                                                                                                                                                                                                                                                                                                                                                                                                                                                                                                                                                                                                                                                                                                                                                                                                                      | h.lat'].self.lat):                                                                                                                                  |
| writejs1Darray(fid, [modelname '.mes                                                                                                                                                                                                                                                                                                                                                                                                                                                                                                                                                                                                                                                                                                                                                                                                                                                                                                                                                                                                                                                                                                                                                                                                                                                                                                                                                                                                                                                                                                                                                                                                                                                                                                                                                                                                                                                                                                                                                                                                                                                                                                                                                                                                                                                                                                                                                                             | h.long'],self.long);                                                                                                                                |
| unitariadaula] a CEi d. Emadal nama I. maala                                                                                                                                                                                                                                                                                                                                                                                                                                                                                                                                                                                                                                                                                                                                                                                                                                                                                                                                                                                                                                                                                                                                                                                                                                                                                                                                                                                                                                                                                                                                                                                                                                                                                                                                                                                                                                                                                                                                                                                                                                                                                                                                                                                                                                                                                                                                                                     | onsall solf onsalt                                                                                                                                  |
| writejsdouble(tid,[modelname .mesh                                                                                                                                                                                                                                                                                                                                                                                                                                                                                                                                                                                                                                                                                                                                                                                                                                                                                                                                                                                                                                                                                                                                                                                                                                                                                                                                                                                                                                                                                                                                                                                                                                                                                                                                                                                                                                                                                                                                                                                                                                                                                                                                                                                                                                                                                                                                                                               | epsg _,setT.epsg/,                                                                                                                                  |
| <pre>end % }}}</pre> end                                                                                                                                                                                                                                                                                                                                                                                                                                                                                                                                                                                                                                                                                                                                                                                                                                                                                                                                                                                                                                                                                                                                                                                                                                                                                                                                                                                                                                                                                                                                                                                                                                                                                                                                                                                                                                                                                                                                                                                                                                                                                                                                                                                                                                                                                                                                                                                         |                                                                                                                                                     |
| end % }}
end                                                                                                                                                                                                                                                                                                                                                                                                                                                                                                                                                                                                                                                                                                                                                                                                                                                                                                                                                                                                                                                                                                                                                                                                                                                                                                                                                                                                                                                                                                                                                                                                                                                                                                                                                                                                                                                                                                                                                                                                                                                                                                                                                                                                                                                                                                                                                                                                  | epsg ],settepsg),                                                                                                                                   |
| end % }}
end % }}
end
end
MESH2D class definition                                                                                                                                                                                                                                                                                                                                                                                                                                                                                                                                                                                                                                                                                                                                                                                                                                                                                                                                                                                                                                                                                                                                                                                                                                                                                                                                                                                                                                                                                                                                                                                                                                                                                                                                                                                                                                                                                                                                                                                                                                                                                                                                                                                                                                                                                                                                                    | , cpsg ], sett. cpsg),                                                                                                                              |
| end % }}
end % }}
end
end
end
MESH2D class definition
function mesh2d () {                                                                                                                                                                                                                                                                                                                                                                                                                                                                                                                                                                                                                                                                                                                                                                                                                                                                                                                                                                                                                                                                                                                                                                                                                                                                                                                                                                                                                                                                                                                                                                                                                                                                                                                                                                                                                                                                                                                                                                                                                                                                                                                                                                                                                                                                                                                     | , cpsg ], sett. cpsg),                                                                                                                              |
| <pre>write_isaduble(rtd, imodelnamemesh end % }} end end //MESH2D class definition function mesh2d () {     //methods     + 10 lines: this setdefmultparameta </pre>                                                                                                                                                                                                                                                                                                                                                                                                                                                                                                                                                                                                                                                                                                                                                                                                                                                                                                                                                                                                                                                                                                                                                                                                                                                                                                                                                                                                                                                                                                                                                                                                                                                                                                                                                                                                                                                                                                                                                                                                                                                                                                                                                                                                                                             | rs = function Ω/                                                                                                                                    |
| <pre>write_isaduble(rtd, imodelname .mesh end % }} end wite_isaduble(rtd, imodelname .mesh end wite_isaduble(rtd, imodelname .mesh end wite_isaduble(rtd, imodelname .mesh end wite_isaduble(rtd, imodelname .mesh wite_isaduble(rtd, imodelname .mesh</pre>                                                                                                                                                                                                                                                                             | rs = function Q{                                                                                                                                    |
| <pre>write_isaduble(rtd, imodelnamemesh end % }} end {/MESH2D class definition function mesh2d () {     //methods + 10 lines: this.setdefaultparamete + 3 lines: this.classname = functi + 3 lines: this.classname = functio </pre>                                                                                                                                                                                                                                                                                                                                                                                                                                                                                                                                                                                                                                                                                                                                                                                                                                                                                                                                                                                                                                                                                                                                                                                                                                                                                                                                                                                                                                                                                                                                                                                                                                                                                                                                                                                                                                                                                                                                                                                                                                                                                                                                                                              | rs = function (){                                                                                                                                   |
| <pre>write_isaduble(rid, imodelname .mesh
end % }}
end
end
//MESH2D class definition
function mesh2d () {
//methods
+ 10 lines: this.setdefaultparamete
+ 3 lines: this.classname = functi
+ 3 lines: this.dimension = functi
+ 3 lines: this.dimension = functi
+ 3 lines: this.dimension = functi</pre>                                                                                                                                                                                                                                                                                                                                                                                                                                                                                                                                                                                                                                                                                                                                                                                                                                                                                                                                                                                                                                                                                                                                                                                                                                                                                                                                                                                                                                                                                                                                                                                                                                                                                                                                                                                                                                                                                                                                                                                                                                                            | <pre>ins = function (){</pre>                                                                                                                       |
| <pre>write_isaduble(rtd, imodelname .mesh
end % }}
end
end
//MESH2D class definition
function mesh2d () {
//methods
+ 10 lines: this.setdefaultparamete
+ 3 lines: this.classname = functi
+ 3 lines: this.dimension = functi
+</pre>                                                                                                                                             | rs = function (){                                                                                                                                   |
| <pre>write_isouble(rtd, imodelname imesh
end % }}
end
//MESH2D class definition
function mesh2d () {
//methods
+ 10 lines: this.setdefaultparamete
+ 3 lines: this.classname = functi
+ 3 lines: this.domaintype=functio
+ 3 lines: this.domentype = funct
this.menshall=function(md,fid) { // {{
WriteData(fid, 'enum',DomainTypeEnum()</pre>                                                                                                                                                                                                                                                                                                                                                                                                                                                                                                                                                                                                                                                                                                                                                                                                                                                                                                                                                                                                                                                                                                                                                                                                                                                                                                                                                                                                                                                                                                                                                                                                                                                                                                                                                                                                                                                                                                                                                                                                                       | <pre>srs = function (){</pre>                                                                                                                       |
| <pre>writeData(fid,'enum', MashElementume') .mesh
end % }}
end
//MESHZD class definition
function mesh2d () {
//methods
+ 3 lines: this.classname = functi
+ 3 lines: this.domaintype=functio
+ 3 lines: this.domaintype=functio
+ 3 lines: this.elementupe = functi
+ 3 lines: this.elementupe = function
function.method(fid,'enum', DomainTypeEnum()
WriteData(fid,'enum', MashElementupe)</pre>                                                                                                                                                                                                                                                                                                                                                                                                                                                                                                                                                                                                                                                                                                                                                                                                                                                                                                                                                                                                                                                                                                                                                                                                                                                                                                                                                                                                                                                                                                                                                                                                                                                                                                                                                                                                                                      | <pre>rs = function (){</pre>                                                                                                                        |
| <pre>writeData(fid, 'enum', MeshElementtypeE</pre>                                                                                                                                                                                                                                                                                                                                                                                                                                                                                                                                                                                                                                                                                                                                                                                                                                                                                                                                                                                                                                                                                                                                                                                                                                                                                                                                                                                                                                                                                                                                                                                                                                                                                                                                                                                                                                                                                                                                                                                                                                                                                                                                                                                                                                                                                                                                                               | <pre>irs = function (){</pre>                                                                                                                       |
| <pre>write_isduble(rid, imdefinite imesi
end % }}
end
function mesh2d () {
//methods
+ 10 lines: this.setdefaultparamete
+ 3 lines: this.classname = functi
+ 3 lines: this.domaintype=functi
+ 3 lines: this.domaintype=functi
+ 3 lines: this.elementtype = funct
this.marshall=function(md,fid) { //{{
WriteData(fid, 'enum',DomainTypeEnumC)
WriteData(fid, 'enum',DomainTypeEnumC)
WriteData(fid, 'enum',MeshElementtypeE
WriteData(fid, 'object',this, 'class','
WriteData(fid, 'object',this, 'class','</pre>                                                                                                                                                                                                                                                                                                                                                                                                                                                                                                                                                                                                                                                                                                                                                                                                                                                                                                                                                                                                                                                                                                                                                                                                                                                                                                                                                                                                                                                                                                                                                                                                                                                                                                                                                                                                                                 | <pre>ins = function (){</pre>                                                                                                                       |
| <pre>writeData(fid,'enum',MeshZEnumC,'idas','
writeData(fid,'enum',MeshZEnumC,'idas','
WriteData(fid,'enum',MeshZEnumC,'idas','
WriteData(fid,'enum',MeshZEnumC,'idas','
WriteData(fid,'enum',MeshZEnumC,'idas','
WriteData(fid,'enum',MeshZEnumC,'idas','
WriteData(fid,'enum',MeshZEnumC,'idas','
WriteData(fid,'enum',MeshZEnumC,'idas','
WriteData(fid,'enum',MeshZEnumC,'idas','
WriteData(fid,'enum',MeshZEnumC,'idas','
WriteData(fid,'enum',MeshZEnumC,'idas','
WriteData(fid,'enum',MeshZEnumC,'idas','
WriteData(fid,'enum',MeshZEnumC,'idas','
WriteData(fid,'enum',MeshZEnumC,'idas','
WriteData(fid,'enum',MeshZEnumC,'idas',''
WriteData(fid,'enum',MeshZEnumC,'idas',''
WriteData(fid,'enum',MeshZEnumC,'idas',''
WriteData(fid,'enum',MeshZEnumC,'idas',''
WriteData(fid,'enum',MeshZEnumC,'idas',''
WriteData(fid,'enum',MeshZEnumC,'idas',''
WriteData(fid,'enum',MeshZEnumC,'idas',''
WriteData(fid,'enum',MeshZEnumC,'idas',''
WriteData(fid,'enum',MeshZEnumC,'idas',''
WriteData(fid,'enum',MeshZEnumC,'idas',''
WriteData(fid,'enum',MeshZEnumC,''
WriteData(fid,'enum',MeshZEnumC,''
WriteData(fid,'enum',MeshZEnumC,''
WriteData(fid,'enum',MeshZEnumC,''
WriteData(fid,'enum',MeshZEnumC,''
WriteData(fid,'enum',MeshZEnumC,''
WriteData(fid,'enum',MeshZEnumC,''
WriteData(fid,'enum',MeshZEnumC,''
WriteData(fid,'enum',MeshZEnumC,''
WriteData(fid,'enum',MeshZEnumC,''
WriteData(fid,'enum',MeshZEnumC,''
WriteData(fid,'enum',MeshZEnumC,''
WriteData(fid,'enum',MeshZEnumC,''
WriteData(fid,'enum',MeshZEnumC,''
WriteData(fid,'enum',MeshZEnumC,''
WriteData(fid,'enum',MeshZEnumC,''
WriteData(fid,'enum',MeshZEnumC,''
WriteData(fid,''
WriteData(fid,''
WriteData(fid,''
WriteData(fid,''
WriteData(fid,''
WriteData(fid,''
WriteData(fid,''
WriteData(fid,''
WriteData(fid,''
WriteData(fid,''
WriteData(fid,''
WriteData(fid,''
WriteData(fid,''
WriteData(fid,''
WriteData(fid,''
WriteData(fid,''
WriteData(fid,''
WriteData(fid,''
WriteData(fid,''
WriteData(fid,''
WriteData(fid,''
WriteData(fid,''
WriteData(fid,''
WriteData(fid,''
WriteData(fid,''
WriteData(fid,''
WriteData(</pre> | <pre>rs = function (){</pre>                                                                                                                        |
| <pre>writeData(fid,'object',this,'class',' writeData(fid,'object',this,'class',' </pre>                                                                                                                                                                                                                                                                                                                                                                                                                                                                                                                                                                                                                                                                                                                                                                                                                                                                                                                                                                                                                                                                                                                                                                                                                                                                                                                                                                                                                                                                                                                                                                                                                                                                                                                                                                                                                                                                                                                                                                                                                                                                                                                                                                                                                                                                                                                          | <pre>ins = function (){</pre>                                                                                                                       |
| <pre>writeData(fid,'object',this,'class',' WriteData(fid,'object',this,'class',' WriteData(fid,'object',this,'class',') WriteData(fid,'object',this,'class',') WriteData(fid,'object',this,'class',') WriteData(fid,'object',this,') </pre>                                                                                                                                                                                                                                                                                                            | <pre>rs = function (){</pre>                                                                                                                        |
| <pre>writeJsaduble(rtd, imodelname imesr
end % }}
end
MESHZD class definition
function mesh2d () {
//methods
+ 10 lines: this.classname = functi
+ 3 lines: this.classname = functi
+ 3 lines: this.dimension = functi
+ 3 lines: this.dimension = functi
+ 3 lines: this.elementtype = func
this.marshall=function(md,fid) { //{{
WriteDatac(fid, 'enum',DomainDimensionE
WriteDatac(fid, 'enum',DomainDimensionE
WriteDatac(fid, 'enum',DomainDimensionE
WriteDatac(fid, 'object',this, 'class','
WriteDatac(fid, 'bject',this, 'class', '
WriteDatac(fid, 'bject',this, 'bject</pre>                                                                                   | <pre>irs = function (){</pre>                                                                                                                       |
| <pre>writeJsaduble(rtd, imodelname imesr
end % }}
end
//MESHZD class definition
function mesh2d () {
//methods
+ 10 lines: this.classname = functi
+ 3 lines: this.classname = functi
+ 3 lines: this.dimension = functi
+ 3 lines: this.dimension = functi
+ 3 lines: this.elementtype = func
this.marshall=function(md,fid) { //{{
WriteData(fid, 'enum', DomainDimensionE
WriteData(fid, 'enum', DomainDimensionE
WriteData(fid, 'enum', MeshElementtypeE
WriteData(fid, 'enum', MeshElementtypeE
WriteData(fid, 'object', this, 'class', '
WriteData(fid, 'object', this, 'class', '
HorieData(fid, 'object', this, 'class', '
WriteData(fid, 'object', this, 'class', '
WriteData(fid, 'object', this, 'class', '
WriteData(fid, 'object', this, 'class', '
HorieData(fid, 'object', this, 'cla</pre>                                                                           | <pre>irs = function (){</pre>                                                                                                                       |
| <pre>writeJsaduble(rtd, imodelname imesr
end % }}
end
md
MESHZD class definition
function mesh2d () {
//methods
+ 10 lines: this.classname = functi
+ 3 lines: this.classname = functi
+ 3 lines: this.dimension = functi
+ 3 lines: this.dimension = functi
+ 3 lines: this.elementtype = func
this.marshall=function(md,fid) { //{{
WriteDatac(fid, 'enum',DomainDimensionE
WriteDatac(fid, 'enum',DomainDimensionE
WriteDatac(fid, 'object',this, 'class','
WriteDatac(fid, 'bject',this, 'class','
WriteDatac(fid, 'bject',this, 'class','
WriteDatac(fid, 'bject',this, 'class','
WriteDatac(fid, 'bject',this, 'class','
WriteDatac(fid, 'bject',this, 'class','
WriteDatac(fid, 'bject',this, 'class', '
WriteDatac(fid, 'bject',this, 'c</pre>                                                                   | <pre>irs = function (){</pre>                                                                                                                       |
| <pre>writeJsaduble(Fld, [modelnamemesh
end % }}
end
md
MESHZD class definition
function mesh2d () {
//methods
+ 10 lines: this.classname = functi
+ 3 lines: this.classname = functi
+ 3 lines: this.dimension = functi
+ 3 lines: this.elementtype = func
this.marshall=function(md,fid) { //{{
WriteData(fid, 'enum', DomainDimensionE
WriteData(fid, 'enum', DomainDimensionE
WriteData(fid, 'object',this, 'class','
WriteData(fid, 'object',this, 'class','
WriteData(fid, 'object',this, 'class','
WriteData(fid, 'object',this, 'class','
WriteData(fid, 'object',this, 'class','
WriteData(fid, 'object',this, 'class','
//
WriteData(fid, 'object',this, 'class','
//
WriteData(fid, 'bject',this, 'class','
//
//
//
//
//
//
//
//
//
//
//
//
//</pre>                                                                                                                                                                                                                                                                                                                                                                                                                                                                                                                                                                                                                                                                                                                                                                                                                                                                                                                                                                                                                                                                                                                                                                                                                                                                                                                                                                          | <pre>irs = function (){</pre>                                                                                                                       |
| <pre>writeJsaduble(Fld, [modelnumemesh
end % }}
end
md
//MESHZD class definition
function mesh2d () {
//methods
+ 10 lines: this.classname = functi
+ 3 lines: this.classname = functi
+ 3 lines: this.dimension = functi
+ 3 lines: this.dimension = functi
+ 3 lines: this.elementype = func
this.marshall=function(md,fid) { //{{
WriteData(fid, 'enum', DomainDimensionE
WriteData(fid, 'enum', DomainDimensionE
WriteData(fid, 'object',this, 'class','
WriteData(fid, 'bject',this, 'class','
WriteData(fid, 'bj</pre>                                                          | <pre>ins = function (){</pre>                                                                                                                       |
| <pre>write_isouble(rid, imodelnume imesh
end % }}
end
end
//MESHZD class definition
function mesh2d () {
//methods
+ 10 lines: this.setdefaultparamete
+ 3 lines: this.classname = functi
+ 3 lines: this.domaintype=functio
+ 3 lines: this.dimension = functi
+ 3 lines: this.dementtype = func
this.marshall=function(md, fid) { //{{
WriteData(fid, 'enum', DomainDimensionE
WriteData(fid, 'enum', DomainDimensionE
WriteData(fid, 'object', this, 'class','
WriteData(fid, 'object', this, 'class', '
WriteData(fid, 'object', this, 'class', '
WriteDa</pre>                                                                   | <pre>ins = function (){</pre>                                                                                                                       |
| <pre>write_isouble(rid, imodelnume imesi
end % }}
end
end
//MESHZD class definition
function mesh2d () {
//methods
+ 10 lines: this.setdefaultparamete
+ 3 lines: this.classname = functi
+ 3 lines: this.domaintype=functio
+ 3 lines: this.dimension = functi
+ 3 lines: this.dementtype = funct
this.marshall=function(md, fid) { //{{
WriteData(fid, 'enum', MeshElementtypeE
WriteData(fid, 'enum', MeshElementtypeE
WriteData(fid, 'object', this, 'class','
WriteData(fid, 'object', this, 'class', '
WriteData(fid, '</pre>                                                                   | <pre>rs = function (){</pre>                                                                                                                        |
| <pre>write_isouble(rid, imodelnume imesr
end % }}
end
end
//MESHZD class definition
function mesh2d () {
//methods
+ 10 lines: this.setdefaultparamete
+ 3 lines: this.classname = functi
+ 3 lines: this.domaintype=functio
+ 3 lines: this.dimension = functi
+ 3 lines: this.elementtype = func
this.marshall=function(md, fid) { //i{{
WriteData(fid, 'enum', DomainTypeEnum()
WriteData(fid, 'enum', MeshElementtypeE
WriteData(fid, 'object', this, 'class','
WriteData(fid, 'object', this, 'class', '
WriteData(fid, 'object', this, 'class', '
WriteData(</pre>                                                               | <pre>rs = function (){</pre>                                                                                                                        |
| <pre>writeJsaduble(rtd, [modelnumemesn
end % }}
end
/MESHZD class definition
function mesh2d () {
//methods
+ 10 lines: this.setdefaultparamete
+ 3 lines: this.classname = functi
+ 3 lines: this.domaintype=functi
+ 3 lines: this.domaintype=functi
+ 3 lines: this.elementtype = func
this.morshall=function(md,fid) { // {{
WriteData(fid, 'enum',DomainTypeEnum()
WriteData(fid, 'enum',DomainTypeEnum()
WriteData(fid, 'enum',MeshElementtypeE
WriteData(fid, 'object',this, 'class','
WriteData(fid, 'object',this, 'class', '
WriteData(fid, 'object',this, 'class', '
WriteData(fi</pre>                                                                      | <pre>rs = function (){</pre>                                                                                                                        |
| <pre>write_isouble(rid, imodelnume imesr
end % }}
end
/MESHZD class definition
function mesh2d () {
//methods
+ 10 lines: this.setdefaultparamete
+ 3 lines: this.classname = functi
+ 3 lines: this.domaintype=functio
+ 3 lines: this.idmension = functi
+ 3 lines: this.idmentype=functio
this.menshall=function(md,fid) { // {{
WriteData(fid, 'enum',DomainTypeEnum()
WriteData(fid, 'enum',MeshElementtypeE
WriteData(fid, 'object',this,'class','
WriteData(fid, 'object',this,'class', '
WriteData(fid, 'object',this,'class', '
WriteData(fid, 'object',this,'class', '
WriteData(fid, 'object',this,'class', '
WriteData(fid, 'object',thi</pre>                                                               | <pre>rs = function (){</pre>                                                                                                                        |
| <pre>writejsdoble(rtd, imodelnume imesr
end % }}
end
md
MitSHZD class definition
function mesh2d () {
//methods
+ 10 lines: this.setdefaultparamete
+ 3 lines: this.classname = functi
+ 3 lines: this.domaintype=functi
+ 3 lines: this.domaintype=functi
+ 3 lines: this.elementype = functi
+ 3 lines: this.elementype = functi
+ 3 lines: this.elementype=function
WriteData(fid, 'enum',DomainTypeEnumC)
WriteData(fid, 'enum',MeshElementtypeE
WriteData(fid, 'object',this, 'class','
WriteData(fid, 'object',this, 'class', '
WriteData(fid, 'object', this, 'class', '
Write</pre>                                                                   | <pre>rs = function (){</pre>                                                                                                                        |
| <pre>writejsdoble(rtd, [modelnamemesn
end % }}
end
methods
=</pre>                                                                                                                                                                                                                                                                                                                                                                                                                                                                                                                                                                                                                                                                                                                                                                                                                                                                                                                                                                                                                                                                                                                                                                                                                                                                                                                                                                                                                                                                                                                                                                                                                                                                                                                                                                                                                                                                                                                                                                                                                                                                                                                                                                                                                                                                                                                               | <pre>rs = function (){</pre>                                                                                                                        |
| <pre>write_isouble(rid, imodelname imesi
end % }}
end
md
End
md
methods
=</pre>                                                                                                                                                                                                                                                                                                                                                                                                                                                                                                                                                                                                                                                                                                                                                                                                                                                                                                                                                                                                                                                                                                                                                                                                                                                                                                                                                                                                                                                                                                                                                                                                                                                                                                                                                                                                                                                                                                                                                                                                                                                                                                                                                                                                                                                                                                      | <pre>rs = function (){</pre>                                                                                                                        |

Figure 1. Line by line comparison of the code behind the mesh2d class, within the MATLAB ISSM API (upper frame) and the JavaScript ISSM API (lower frame). Routines followed by a dashed line have been folded for ease of reading. 12

---

## Editor Comment (EC1) · D. Goldberg (Editor) · 4 Feb 2017

**General comments**

This manuscript describes a JavaScript version of the Ice Sheet System Model that enables users to interact with the model via a web browser. My understanding is that users are able to set up simulations via the web interface and either run them directly on the web server or indirectly via through a job that is automatically submitted to a cluster or supercomputer. In either case, the results are displayed interactively using the browser's 3D graphics capabilities. The concept is that this approach will make ISSM more accessible to non-experts without compromising capabilities or requiring separate development of the core ISSM capabilities for this purpose.

The paper is well written and only some fairly minor revisions are required to make it ready for publication, as outlined in my specific comments below.

The other 2 reviewers expresses a certain amount of confusion about what the intended audience for the JavaScript representation might be. While I felt that was fairly clear to me, I can appreciate that further explanation would be useful. The translation of so much of ISSM to JavaScript must have been quite a tour de force, something that other ice sheet models are unlikely to undertake unless it is very clear what the benefits of developing (and maintaining) a JavaScript version of their models will be.

I was very impressed with the online examples on the VESL. While I share Reviewer 2's view that users are currently restricted to an extent that the example results could have been pre-run and cached, I appreciate the potential versatility that could be built into this system, quickly making pre-caching results completely impractical. I'm pleasantly surprised with the quick turnaround from setup to viz that this system allows (presumably because the examples are relatively light weight, but even so). I look forward to seeing how the VESL develops.

**Specific comments**

l. 62: It is probably a question of reference frame, but do you perhaps mean "downloading" instead of "uploading"? My picture was of the server automatically downloading the data it needs to run the simulation, whereas uploading seems like something the user has to do manually. Perhaps I'm just unclear on what exactly is being moved to the server and from where.

l. 119-120: "The basis for representing a model in ISSM is a series of classes (mesh, mask, geometry, settings, toolkits, etc.) that are carried into a global model class." I'm unclear as to what exactly is meant by a "model" at the beginning of this sentence and by "carried into" in the second part. Please explain these two concepts in a bit more detail. My sense is that what you refer to as a "model" here is what I am more familiar with as a model setup or a test case or something like that. My guess is that "carried into" might mean that the global class contains instances of each of the other classes, but I really am not clear on what is meant.

Supplement:

I tried to compile ISSM with JavaScript under Ubuntu 16.4 but was not successful. I realize the supplement states that only MacOSX is currently supported, but I decided to give it a try. When I run the configure command as given in the supplement, the command hangs after printing the line:

```
checking whether the em++ linker (/usr/bin/ld -m elf_x86_64) supports shared
libraries... yes
```
I wasn't able to figure out what command is being executed that causes the hang.

Anyway, that is to say that I think probably more support will be needed to get the JavaScript version of ISSM up and running on more platforms.

**Typographic corrections**

l. 4-5: Here and elsewhere in the manuscript, I think "pre and post-processing" should be "pre- and post-processing" with two hyphens.
l. 5: I think "i.e." should probably be "e.g." since other tools such as IDL are also sometimes used for this purpose, as pointed out later in the manuscript.
l. 5-6: "non specialists" should be "non-specialists"

l. 139: "In addition to the classes representation in JavaScript..." I think "classes" should just be "class".

l. 189 and 191: "respective to" should probably be "depending on"

l. 230: "all without loss of the physical representation of processes nor scalability" here, "nor" should be "or".

l. 237: "The first simulations pertain to the simulation of glacier flow..." Presumably "simulations" could be changed to "setups" or similar to avoid redundant use of the word "simulation"?

Fig. 1: I would suggest changing this figure to be a listing instead. If you prefer to keep it as a figure, I would suggest changing the color scheme to be dark text on a light background to make it more print-friendly and easier on the eyes (especially at this rather small font size).

l. 331: There seems to be a stray semicolon here, right before before Listing 2.

Supplement:
l. 7-8: "...JavaScript support turned on lies in the specific configuration..." You probably want to keep only one or the other of "turned on" or "lies in".